# A Bounded Measure for Estimating the Benefit of Visualization (Part I): Theoretical Discourse and Conceptual Evaluation

**DOI:** 10.3390/e24020228

**Published:** 2022-01-31

**Authors:** Min Chen, Mateu Sbert

**Affiliations:** 1Oxford e-Research Centre (OeRC), Department of Engineering Science, University of Oxford, Oxford OX1 3QG, UK; 2Department of Informàtica i Matemàtica Aplicada, University of Girona, 17071 Girona, Spain; mateu@ima.udg.edu

**Keywords:** information theory, theory of visualization, cost–benefit analysis, divergence measure, benefit of visualization, human knowledge in visualization, abstraction, deformation, volume visualization, metro map

## Abstract

Information theory can be used to analyze the cost–benefit of visualization processes. However, the current measure of benefit contains an unbounded term that is neither easy to estimate nor intuitive to interpret. In this work, we propose to revise the existing cost–benefit measure by replacing the unbounded term with a bounded one. We examine a number of bounded measures that include the Jenson–Shannon divergence, its square root, and a new divergence measure formulated as part of this work. We describe the rationale for proposing a new divergence measure. In the first part of this paper, we focus on the conceptual analysis of the mathematical properties of these candidate measures. We use visualization to support the multi-criteria comparison, narrowing the search down to several options with better mathematical properties. The theoretical discourse and conceptual evaluation in this part provides the basis for further data-driven evaluation based on synthetic and experimental case studies that are reported in the second part of this paper.

## 1. Introduction

To most of us, it seems rather intuitive that visualization should be accurate, different data values should be visually encoded differently, and visual distortion should be disallowed. However, when we closely examine most (if not all) visualization images, we can notice that inaccuracy is ubiquitous. The two examples in Figure 1 evidence the presence of such inaccuracy. In volume visualization, when a pixel is used to depict a set of voxels along a ray, many different sets of voxel values may result in the same pixel color. In a metro map, a variety of complex geographical paths may be distorted and depicted as a straight line. Since there is little doubt that volume visualization and metro maps are useful, some “inaccurate” visualization must be beneficial.

In terms of information theory, the types of inaccuracy featured in Figure 1 are different forms of information loss (or many-to-one mapping). Chen and Golan proposed an information-theoretic measure [1] for analyzing the cost–benefit of data intelligence workflows. It enables us to consider the positive impact of information loss (e.g., reducing the cost of storing, processing, displaying, perceiving, and reasoning about the information) as well as its negative impact (e.g., being mislead by the information). The measure provides a concise explanation about the benefit of visualization because visualization and other data intelligence processes (e.g., statistics and algorithms) all typically cause information loss and visualization allows human users to reduce the negative impact of information loss effectively using their knowledge.

The mathematical formula of the cost–benefit ratio features a term based on the Kullback–Leibler (KL) divergence [2] for measuring the potential distortion of a user or a group of users in reconstructing the information that may have been lost or distorted during a visualization process. The cost–benefit ratio instigates that a user with more knowledge about the source data and its visual representation is likely to suffer less distortion. While using the KL-divergence is mathematically intrinsic for measuring the potential distortion, its unboundedness property has some undesirable consequences. The simplest phenomenon of making a false representation (i.e., always displaying 1 when a binary value is 0 or always 0 when it is 1) happens to be a singularity condition of the KL-divergence. The amount of distortion measured by the KL-divergence often has many more bits than the entropy of the information space itself. This is not intuitive to interpret and hinders practical applications.

In this two-part paper, we propose to replace the KL-divergence with a bounded term. In the first part, we first confirm the boundedness is a necessary property. We then conduct multi-criteria decision analysis (MCDA) [3] to compare a number of bounded measures, which include the Jensen–Shannon (JS) divergence [4], its square root, and a new divergence measure $D_{\mathrm{new}}^{k}$ (including its variations) formulated as part of this work. We use visual analysis to aid the observation of the mathematical properties of these candidate measures, narrowing down from eight options to five. In the second part of this paper [5], we use synthetic and experimental case studies to to instantiate values that may be returned by the five options. It also explores the relationship between measuring the benefit of visualization and measuring the viewers’ knowledge used during visualization.

The search for the best way to measure the benefit of visualization will likely entail a long journey. The main aim of this work is to initiate this endeavor. The main technical contributions of this two-part paper include:Identifying a shortcoming of using the KL-divergence in the information-theoretic measure proposed by Chen and Golan [1] and evidencing the shortcoming using practical examples (Parts I and II);Presenting a theoretical discourse to justify the use of a bounded measure for finite alphabets (Part I);Proposing a new bounded divergence measure, while studying existing bounded divergence measures (Part I);Analyzing nine candidate measures using seven criteria reflecting desirable conceptual or mathematical properties, and narrowing the nine candidate measures to six measures (Part I);Conducting several case studies for collecting instances for evaluating the remaining six candidate measures (Part II);Demonstrating the uses of the cost–benefit measurement to estimate the benefit of visualization in practical scenarios and the human knowledge used in the visualization processes (Part II);Discovering a new conceptual criterion that a divergence measure is a summation of the entropic values of its components, which is useful in analyzing and visualizing empirical data (Part II);Offering a recommendation to revise the information-theoretic measure proposed by Chen and Golan [1] based on multi-criteria decision analysis (Parts I and II).

## 2. Related Work

Claude Shannon’s landmark article in 1948 [6] signifies the birth of information theory. It has been underpinning the fields of data communication, compression, and encryption since. As a mathematical framework, information theory provides a collection of useful measures, many of which, such as Shannon entropy [6], cross entropy [7], mutual information [7], and Kullback–Leibler divergence [2] are widely used in applications of physics, biology, neurology, psychology, and computer science (e.g., visualization, computer graphics, computer vision, data mining, machine learning), and so on. In this work, we also consider Jensen-Shannon divergence [4] in detail.

Information theory has been used extensively in visualization [8]. It has enabled many applications in visualization, including scene and shape complexity analysis by Feixas et al. [9] and Rigau et al. [10], light source placement by Gumhold [11], view selection in mesh rendering by Vázquez et al. [12] and Feixas et al. [13], attribute selection by Ng and Martin [14], view selection in volume rendering by Bordoloi and Shen [15], and Takahashi and Takeshima [16], multi-resolution volume visualization by Wang and Shen [17], focus of attention in volume rendering by Viola et al. [18], feature highlighting by Jänicke and Scheuermann [19,20], and Wang et al. [21], transfer function design by Bruckner and Möller [22], and Ruiz et al. [23,24], multi-modal data fusion by Bramon et al. [25], isosurface evaluation by Wei et al. [26], measuring observation capacity by Bramon et al. [27], measuring information content by Biswas et al. [28], proving the correctness of “overview first, zoom, details-on-demand” by Chen and Jänicke [29] and Chen et al. [8], and confirming visual multiplexing by Chen et al. [30].

Ward first suggested that information theory might be an underpinning theory for visualization [31]. Chen and Jänicke [29] outlined an information-theoretic framework for visualization, and it was further enriched by Xu et al. [32] and Wang and Shen [33] in the context of scientific visualization. Chen and Golan proposed an information-theoretic measure for analyzing the cost–benefit of visualization processes and visual analytics workflows [1]. It was used to frame an observation study showing that human developers usually entered a huge amount of knowledge into a machine learning model [34]. It motivated an empirical study confirming that knowledge could be detected and measured quantitatively via controlled experiments [35]. It was used to analyze the cost–benefit of different virtual reality applications [36]. It formed the basis of a systematic methodology for improving the cost–benefit of visual analytics workflows [37]. It survived qualitative falsification by using arguments in visualization [38]. It offered a theoretical explanation of “visual abstraction” [39]. It provided a theoretical basis to a design space that was structured according to different ways of “losing information” in origin-destination data visualization [40]. The work reported in this paper continues the path of theoretical developments in visualization [41], and is intended to improve the original cost–benefit formula [1], in order to make it a more intuitive and usable measurement in practical visualization applications.

The information-theoretic measure proposed by Chen and Golan [1] can be applied to a variety of processes for transforming some input data to some output data [42]. These include machine-centric processes (e.g., computing statistical measures, importance sampling, feature extraction, dimensionality reduction, etc.) and human-centric processes (e.g., data visualization, human-computer interaction, written communication, human cognition, etc.). In this work, we focus on processes of data visualization.

## 3. Overview and Motivation

A short introduction to information-theoretic cost–benefit analysis can be found in an arXiv report [43]. For self-containment, we provide a brief overview to accompany our description of the problem that motivated this work.

Visualization is useful in most data intelligence workflows, but the usefulness is not universally true because the effectiveness of visualization is usually data-, user-, and task-dependent. The cost–benefit ratio proposed by Chen and Golan [1] captures the essence of such dependency. Below is the qualitative expression of the measure:(1)$$\frac{\mathrm{Benefit}}{\mathrm{Cos}t}=\frac{\mathrm{Alphabet}\;\mathrm{Compression}-\mathrm{Potential}\;\mathrm{Distortion}}{\mathrm{Cos}t}$$

Consider the scenario of viewing some data through a particular visual representation. The term *Alphabet Compression* (AC) measures the amount of information loss due to visual abstraction [39] (or any transformation featuring many-to-one mappings). Since the visual representation is fixed in the scenario, AC is thus largely data-dependent. AC is a positive measure reflecting the fact that visual abstraction must be useful in many cases though it may result in information loss. This apparently counter-intuitive term is essential for asserting why visualization is useful. Note that the term also helps assert the usefulness of statistics, algorithms, and interaction since they all usually cause information loss [37].

The positive implication of the term AC is counterbalanced by the term *Potential Distortion*, while both being moderated by the term *Cost*. The term *Cost* encompasses all costs of the visualization process, including computational costs (e.g., visual mapping and rendering), cognitive costs (e.g., cognitive load), and consequential costs (e.g., impact of errors). As illustrated in Figure 2, increasing AC typically enables the reduction of cost (e.g., in terms of energy, or its approximation such as time or money).

The term *Potential Distortion* (PD) measures the informative divergence between viewing the data through visualization with information loss and reading the data without any information loss. The latter might be ideal but is usually at an unattainable cost except for values in a very small data space (i.e., in a small alphabet as discussed in [1]). As shown in Figure 2, increasing AC typically causes more PD. PD is data-dependent or user-dependent. Given the same data visualization with the same amount of information loss, one can postulate that a user with more knowledge about the data or visual representation usually suffers less distortion. This postulation is a focus of this paper.

Consider the visual representation of a network of arteries in Figure 3. The image was generated from a volume dataset using the maximum intensity projection (MIP) method. While it is known that MIP cannot convey depth information well, it has been widely used for observing some classes of medical imaging data, such as arteries. The highlighted area in Figure 3 shows an apparently flat area, which is a distortion from the actuality of a tubular surface likely with some small wrinkles and bumps. The doctors who deal with such medical data are expected to have sufficient knowledge to reconstruct the reality adequately from the “distorted” visualization, while being able to focus on the more important task of making diagnostic decisions, e.g., about aneurysm.

As shown in some recent works, it is possible for visualization designers to estimate AC, PD, and Cost qualitatively [36,37] and quantitatively [34,35]. It is highly desirable to advance the scientific methods for quantitative estimation, towards the eventual realization of computer-assisted analysis and optimization in designing visual representations. This work focuses on one challenge of quantitative estimation, i.e., how to estimate the benefit of visualization to human users with different knowledge about the depicted data and visual encoding.

Building on the methods of observational estimation [34] and controlled experiment [35], one may reasonably anticipate a systematic method based on a short interview by asking potential viewers a few questions. For example, one may use the question in Figure 3 to estimate the knowledge of doctors, patients, and any other people who may view such a visualization. The question is intended to tease out two pieces of knowledge that may help reduce the potential distortion due to the “flat area” depiction. One piece is about the general knowledge that associates arteries with tube-like shapes. Another, which is more advanced, is about the surface texture of arteries and the limitations of the MIP method.

In the second part of this paper [5], the question in Figure 3 is one of eight questions used in a survey for collecting empirical data for evaluating the bounded measures considered in this paper. As this paper focuses on the theoretical discourse and conceptual evaluation, we use a highly abstracted version of this example to introduce the relevant information-theoretic notations and elaborate the problem statement addressed by this paper.

## 4. Mathematical Notations and Problem Statement

### 4.1. Mathematical Notation

Consider a simplified scenario in Figure 4, where three sequences of voxels are rendered using the MIP method, resulting in three pixel values on the left. Here the three sequence voxels examplifies a volume with $N_{x}\times N_{y}\times N_{Z}$ voxels (illustrated as 1×3×10). Let each voxel value be an 8-bit unsigned integer. In information theory, the possible 256 values are referred to as an *alphabet*, denoted here as $D_{\mathrm{vxl}}=\left\{0,1,2,\ldots ,255\right\}$. The 256 valid values [0,255] are its *letters*. The alphabet is associated with a *probability mass function* (PMF) $P(D_{\mathrm{vxl}})$. The Shannon entropy of this alphabet $H(D_{\mathrm{vxl}})$ measures the average uncertainty and information of the voxel, and is defined as:$$H\left(D_{\mathrm{vxl}}\right)=-\sum_{0}^{255}p_{i}\log_{2}p_{i}\;\mathrm{where}\;p_{i}\in \left[0,1\right],\sum_{0}^{255}p_{i}=1$$
where $p_{i}$ indicates the probability for the voxel to have its value equal to i∈[0,255]. When all 256 values are equally probable (i.e., $\forall i\in \left[0,255\right],p_{i}=1/256$), we have $H(D_{\mathrm{vxl}})=8$ bits. In practice, an application usually deals with a specific type of volume data, the probability of different values may vary noticeably. For example, in medical imaging, a voxel at the boundary of a volume is more likely to have a value indicating an empty space.

The entire volume of $N_{x}\times N_{y}\times N_{Z}$ voxels be defined as a composite alphabet $D_{\mathrm{vlm}}$. Its letters are all valid combinations of voxel values. If the $N_{x}\times N_{y}\times N_{Z}$ voxels are modelled as independent and identically distributed random variables, we have:$$H\left(D_{\mathrm{vml}}\right)=\sum_{k=1}^{M}H\left(D_{\mathrm{vxl},k}\right)=8\times N_{x}\times N_{y}\times N_{Z}\;\mathrm{bits}$$
where $M=N_{x}\times N_{y}\times N_{z}$. For the volume illustrated in Figure 4, $H(D_{\mathrm{vml}})$ would be 240 bits. However, this is the maximum entropy of such a volume. In real world applications, it is very unlikely for the $N_{x}\times N_{y}\times N_{Z}$ voxels to be independent and identically distributed random variables. Although domain experts may not have acquired the ground truth PMF, by measuring a very large corpus of volume data, they have intuitive knowledge as to what may be possible or not. For example, doctors, who handle medical imaging data, do not expect to see a car, ship, aeroplane, or other “weird” objects in a volume dataset. This intuitive and imprecise knowledge about the PMF can explain the humans’ ability to decode visualization featuring some “short comings” such as various visual multiplexing phenomena (e.g., occlusion, displacement, and information omission) [30]. In the second part of this paper [5], we will explore means to measure such ability quantitatively.

Similarly, we can define an alphabet for an pixel $R_{\mathrm{pxl}}$ and a composite alphabet for an image $R_{\mathrm{img}}$. For the example in Figure 4, we assume a simple MIP algorithm that selects the maximum voxel value along each ray, and assigns it as the corresponding pixel as an 8-bit monochromatic value. It is obvious that the potential variation of a pixel is much less than the potential combined variation of all voxels along a ray. Hence in terms of Shannon entropy, most likely $H\left(D_{\mathrm{vlm}}\right)-H\left(D_{\mathrm{img}}\right)⋙0$, indicating significant information loss during the rendering process.

Given an analytical task to be performed through visualization, the analytical decision alphabet A usually contains a small number of letters, such as {*contain artefact X*, *no artefact X*} or {*big*, *medium*, *small*, *tiny*, *none*}. The entropy of A is usually much lower than that of $D_{\mathrm{img}}$, i.e., $H\left(D_{\mathrm{img}}\right)-H\left(A\right)\gg 0$, indicating further information loss during perception and cognition. As discussed in Section 3, this is referred to as *alphabet compression* and is a general trend of all data intelligence workflows. The question is thus about how much the analytical decision was disadvantaged by the loss of information. This is referred to as *potential distortion*.

In the original quantitative formula proposed in [1], the potential distortion is measured using Kullback–Leibler divergence (or KL-divergence) [2]. Given an alphabet Z with two PMFs *P* and *Q*, KL-divergence measures how much *Q* differs from the reference distribution *P*:(2)$$D_{\mathrm{KL}}\left(P\left(Z\right)||Q\left(Z\right)\right)=\sum_{i=1}^{n}p_{i}\left(\log_{2}p_{i}-\log_{2}q_{i}\right)=\sum_{i=1}^{n}p_{i}\log_{2}\frac{p_{i}}{q_{i}}$$
where n=∥Z∥ is the number of letters in the alphabet Z, and $p_{i}$ and $q_{i}$ are the probability values associated with letter $z_{i}\in Z$. $D_{\mathrm{KL}}$ is also measured in bit. Because $D_{\mathrm{KL}}$ is an unbounded measure regardless the maximum entropy of Z, it is easy to relate, quantitatively, the value of potential distortion and that of alphabet compression. This leads to the problem to be addressed in this two-part paper.

**Note**: In this paper, to simplify the notations in different contexts, for an information-theoretic measure, we use an alphabet Z and its PMF *P* interchangeably, e.g., H(P(Z))=H(P)=H(Z). An arXiv report [43] provides a short introduction to the cost–benefit analysis and the relevant mathematical background of information theory, which some readers may find helpful.

### 4.2. Problem Statement

Recall our brief discussion about an analytical task that may be affected by the MIP image in Figure 3 in Section 3. Let us define the analytical task as binary options about whether the “flat area” is actually flat or curved. In other words, it is an alphabet A={curved,flat}. The likelihood of the two options is represented by a probability distribution or probability mass function (PMF) P(A)={1−ϵ,0+ϵ}, where 0<ϵ<1. Since most arteries in the real world are of tubular shapes, one can imagine that a ground truth alphabet $A_{G.T.}$ might have a PMF $P(A_{G.T.})$ strongly in favor of the *curved* option. However, the visualization seems to suggest the opposite, implying a PMF $P(A_{\mathrm{MIP}})$ strongly in favor of the *flat* option. It is not difficult to interview some potential viewers, enquiring how they would answer the question. One may estimate a PMF $P(A_{\mathrm{doctors}})$ from doctors’ answers, and another $P(A_{\mathrm{patients}})$ from patients’ answers.

Table 1 shows two scenarios where different probability data is obtained. The values of PD are computed using the KL-divergence as proposed in [1]. In Scenario 1, without any knowledge, the visualization process would suffer 6.50 bits of potential distortion (PD). As doctors are not fooled by the “flat area” shown in the MIP visualization, their knowledge is worth 6.50 bits. Meanwhile, patients would suffer 1.12 bits of PD on average, their knowledge is worth 5.38=6.50−1.12 bits.

In Scenario 2, the PMFs of $P(A_{G.T.})$ and $P(A_{\mathrm{MIP}})$ depart further away, while $P(A_{\mathrm{doctors}})$ and $P(A_{\mathrm{patients}})$ remain the same. Although doctors and patients would suffer more PD, their knowledge is worth more than that in Scenario 1 (i.e., 13.28−0.05=13.23 bits and 13.28−3.11=10.17 bits respectively).

Similarly, the binary options about whether the “flat area” is actually smooth or not can be defined by an alphabet A={wrinkles−and−bumps,smooth}. Table 2 shows two scenarios about collected probability data. In these two scenarios, doctors exhibit much more knowledge than patients, indicating that the surface texture of arteries is a piece of specialized knowledge.

The above example demonstrates that using the KL-divergence to estimate PD can differentiate the knowledge variation between doctors and patients regarding the two pieces of knowledge that may reduce the distortion due to the “flat area”. When it is used in Equation (Equation 1) in a relative or qualitative context (e.g., [36,37]), the unboundedness of the KL-divergence does not pose an issue.

However, this does become an issue when the KL-divergence is used to measure PD in an absolute and quantitative context. From the two diverging PMFs $P(A_{G.T.})$ and $P(A_{\mathrm{MIP}})$ in Table 1, or $P(B_{G.T.})$ and $P(B_{\mathrm{MIP}})$ in Table 2, we can observe that the smaller ϵ is, the more divergent the two PMFs become and the higher value the PD has. Indeed, consider an arbitrary alphabet $Z=\{z_{1},z_{2}\}$, and two PMFs defined upon Z: P=[0+ϵ,1−ϵ] and Q=[1−ϵ,0+ϵ]. When ϵ→0, we have the KL-divergence $D_{\mathrm{KL}}\left(Q||P\right)\to \infty$.

Meanwhile, the Shannon entropy of Z, H(Z), has an upper bound of 1 bit. It is thus not intuitive or practical to relate the value of $D_{\mathrm{KL}}\left(Q||P\right)$ to that of H(Z). Many applications of information theory do not relate these two types of values explicitly. When reasoning such relations is required, the common approach is to impose a lower-bound threshold for ϵ (e.g., [35]). However, there is yet a consistent method for defining such a threshold for various alphabets in different applications, while preventing a range of small or large values (i.e., [0,σ) or (1−σ,1]) in a PMF is often inconvenient in practice. Indeed, for a binary alphabet with two arbitrary *P* and *Q*, in order to restrict its $D_{\mathrm{KL}}\left(P||Q\right)\leq 1$, one has to set 0.0658≲σ≲0.9342, rendering some 13% of the probability range [0, 1] unusable. In the following section, we discuss several approaches to defining a bounded measure for PD.

## 5. Bounded Measures for Potential Distortion (PD)

Let $P_{i}$ be a process in a data intelligence workflow, $Z_{i}$ be its input alphabet, and $Z_{i+1}$ be its output alphabet. $P_{i}$ can be a human-centric process (e.g., visualization and interaction) or a machine-centric process (e.g., statistics and algorithms). In the original proposal [1], the value of Benefit in EquationEquation 1 is measured using:(3)$$\mathrm{Benefit}=\mathrm{AC}-\mathrm{PD}=H\left(Z_{i}\right)-H\left(Z_{i+1}\right)-D_{\mathrm{KL}}(Z_{i}^{\prime }\left|\right|Z_{i})$$
where H() is the Shannon entropy of an alphabet and $D_{\mathrm{KL}}\left(\right)$ is the KL-divergence of an alphabet from a reference alphabet. AC, which is $H\left(Z_{i}\right)-H\left(Z_{i+1}\right)$, defines the entropic difference between the input and output alphabets. Because the Shannon entropy of an alphabet with a finite number of letters is bounded, AC is also bounded. On the other hand, as discussed in the previous section PD (i.e., $D_{\mathrm{KL}}(Z_{i}^{\prime }\left|\right|Z_{i})$) is unbounded. Although Equation (Equation 3) can be used for relative comparison, it is not quite intuitive in an absolute context, and it is difficult to imagine that the amount of informative distortion can be more than the maximum amount of information available.

Given a divergence or difference measure Δ(α,β), the term *bound* may be used in two different contexts. (i) In the *general context*, the bounds of Δ(α,β) are defined based on all possible α and β values in their generic variable domain (e.g., integer, real, or PMF). (ii) In a *specific* or *conditional context*, the bounds of Δ(α,β) are defined based on possible α and β values subject to a specific condition. For example, if we have a specific condition α,β∈[−1,1]⊂R, it is not unusual to expect a difference measure Δ(α,β) to be bounded. However, as discussed in the previous section, the KL-divergence $D_{\mathrm{KL}}\left(P||Q\right)$, can still be unbounded even if we have a finite alphabet Z and the Shannon entropy measures H(P(Z)) and H(Q(Z)) are bounded.

In this section, we present the unpublished work by Chen and Sbert [44], which reasons mathematically that for alphabets of a finite size, the KL-divergence used in Equation (Equation 3) should ideally be bounded. In their arXiv report, they also outlined a new divergence measure and compare it with a few other bounded measures. Building on the initial comparison by Chen and Sbert in [44], we use visualization in Section 6 to assist the multi-criteria analysis and selection of a bounded divergence measure to replace the KL-divergence used in Equation (Equation 3). In the second part of this paper [5], we will further examine the practical usability of a subset of bounded measures by evaluating them using synthetic and experimental data.

### 5.1. A Conceptual Proof of Boundedness

According to the mathematical definition of $D_{\mathrm{KL}}$ in Equation (Equation 2), $D_{\mathrm{KL}}$ is of course unbounded. We do not in anyway try to prove that this formula is bounded. We are interested in a scenario where an alphabet Z is associated with two PMF, *P* and *Q*, which is very much the scenario of measuring the potential distortion in Equation (Equation 1). We ask a question: is it conceptually necessary for $D_{\mathrm{KL}}$ to yield a unbounded value to describe the divergence between *P* and *Q* in this scenario despite that H(P) and H(Q) are both bounded?

We highlight the word “conceptually” because this relates to the concept about another information-theoretic measure, cross entropy, which is defined as:(4)$$H_{\mathrm{CE}}\left(P,Q\right)=\sum_{i=1}^{n}p_{i}\log_{2}\frac{1}{q_{i}}=\sum_{i=1}^{n}p_{i}\log_{2}\frac{p_{i}}{q_{i}}-\sum_{i=1}^{n}p_{i}\log_{2}p_{i}=D_{\mathrm{KL}}\left(P||Q\right)+H\left(P\right)$$
Conceptually, cross entropy measures the cost of a coding scheme. If a code (i.e., an alphabet Z) has a true PMF *P*, the optimal coding scheme should require only H(P) bits according to Shannon’s source coding theorem [7]. However, if the code designer mistakes the PMF as *Q*, the resulting coding scheme will have $H_{\mathrm{CE}}\left(P,Q\right)$ bits. From Equation (Equation 4), we can observe that the inefficiency is described by the term $D_{\mathrm{KL}}\left(P||Q\right)$. Naturally, we can translate our aforementioned question to: should such inefficiency be bounded if there is a finite number of codewords (letters) in the code (alphabet).

Coding theory has been applied to visualization, e.g., for explaining the efficiency of logarithmic plots in displaying data of a family of skewed PMFs and the usefulness of redundancy in visual design [29]. Here, we focus on proving that $H_{\mathrm{CE}}\left(P,Q\right)$ is conceptually bounded.

Let Z be an alphabet with a finite number of letters, $\left\{z_{1},z_{2},\ldots ,z_{n}\right\}$, and Z is associated with a PMF, *Q*, such that:(5)$$\begin{array}{cc} q(z_{n}) & =\epsilon ,\;(\mathrm{where}\;0<\epsilon <2^{-(n-1)}),\\ q(z_{n-1}) & =\left(1-\epsilon \right)2^{-(n-1)},\\ q(z_{n-2}) & =\left(1-\epsilon \right)2^{-(n-2)},\\ \; & \cdots \\ q(z_{2}) & =\left(1-\epsilon \right)2^{-2},\\ q(z_{1}) & =\left(1-\epsilon \right)2^{-1}+\left(1-\epsilon \right)2^{-(n-1)}. \end{array}$$
When we encode this alphabet using an entropy binary coding scheme [45], we can be assured to achieve an optimal code with the lowest average length for codewords. One example of such a code for the above probability is:(6)$$\begin{array}{cc} z_{1} & :0,\;z_{2}:10,\;z_{3}:110\\ \; & \cdots \\ z_{n-1} & :111\ldots 10\;(\mathrm{with}\;n-2\;“1”s\;\mathrm{and}\;\mathrm{one}\;“0”)\;\\ z_{n} & :111\ldots 11\;(\mathrm{with}\;n-1\;“1”s\;\mathrm{and}\;\mathrm{no}\;“0”)\; \end{array}$$
In this way, $z_{n}$, which has the smallest probability, will always be assigned a codeword with the maximal length of n−1. Entropy coding is designed to minimize the average number of bits per letter when one transmits a “very long” sequence of letters in the alphabet over a communication channel. Here the phrase “very long” implies that the string exhibits the above PMF *Q* (Equation (Equation 5)).

Suppose that Z is actually of PMF *P*, but is encoded as Equation (Equation 6) based on *Q*. The transmission of Z using this code will have inefficiency. As mentioned above, the cost is measured by cross entropy $H_{\mathrm{CE}}\left(P,Q\right)$, and the inefficiency is measured by the term $D_{\mathrm{KL}}\left(P||Q\right)$ in Equation (Equation 4).

Clearly, the worst case is that the letter, $z_{n}$, which was encoded using n−1 bits, turns out to be the most frequently used letter in *P* (instead of the least in *Q*). It is so frequent that all letters in the long string are of $z_{n}$. So the average codeword length per letter of this string is n−1. The situation cannot be worse. Therefore, n−1 is the upper bound of the cross entropy. From Equation (Equation 4), we can also observe that $D_{\mathrm{KL}}\left(P||Q\right)$ must also be bounded since $H_{\mathrm{CE}}\left(P,Q\right)$ and H(P) are both bounded as long as Z has a finite number of letters. Let ${⊤}_{\mathrm{CE}}$ be the upper bound of $H_{\mathrm{CE}}\left(P,Q\right)$. The upper bound for $D_{\mathrm{KL}}\left(P||Q\right)$, ${⊤}_{\mathrm{KL}}$, is thus:(7)$$D_{\mathrm{KL}}\left(P||Q\right)=H_{\mathrm{CE}}\left(P,Q\right)-H\left(P\right)\leq {⊤}_{CE}-\min_{\forall P(Z)}\left(H(P)\right)$$

There is a special case worth noting. In practice, it is common to assume that *Q* is a uniform distribution, i.e., $q_{i}=1/n,\forall q_{i}\in Q$, typically because *Q* is unknown or varies frequently. Hence the assumption leads to a code with an average length equaling $\log_{2}n$ (or in practice, the smallest integer $\geq \log_{2}n$). Under this special (but rather common) condition, all letters in a very long string have codewords of the same length. The worst case is that all letters in the string turn out to the same letter. Since there is no informative variation in the PMF *P* for this very long string, i.e., H(P)=0, in principle, the transmission of this string is unnecessary. The maximal amount of inefficiency is thus $\log_{2}n$. This is indeed much lower than the upper bound ${⊤}_{\mathrm{CE}}=n-1$, justifying the assumption or use of a uniform *Q* in many situations.

A more formal proof of the boundedness of $H_{\mathrm{CE}}\left(P,Q\right)$ and $D_{\mathrm{KL}}\left(P||Q\right)$ for an alphabet with a finite number of letters can be found in Appendix A with more detailed discussions. It is necessary to note again that the discourse in this section and Appendix A does not imply that the KL-divergence is incorrect. Firstly, the KL-divergence applies to both discrete probability distributions (PMFs) and continuous distributions. Secondly, the KL-divergence is one of the many divergence measures found in information theory, and a member of the huge collection of statistical distance or difference measures. There is no simply answer as to which measure is correct and incorrect or which is better. We therefore should not over-generalize the proof to undermine the general usefulness of the KL-divergence.

### 5.2. Existing Candidates of Bounded Measures

In practical applications, numerical approximation is commonly used to bound KL-divergence by setting a small value 0<ϵ<0.5 and adjusting probability values in a PMF to ensure all ϵ≤p≤1−ϵ. While numerical approximation may provide a bounded KL-divergence, it is not easy to determine the value of ϵ and it is difficult to ensure everyone to use the same ϵ for the same alphabet or comparable alphabets. For a small alphabet, ϵ has to be a fairly large value, reducing the probability range noticeably. For example, a binary alphabet has maximum Shannon entropy 1 bit. One would need to set an ϵ>0.22 in order bound any $D_{\mathrm{KL}}$ for this alphabet within [0, 1]. It is therefore desirable to consider bounded measures that may be used in place of $D_{\mathrm{KL}}$.

Jensen-Shannon divergence is such a measure:(8)$$\begin{array}{cc} D_{\mathrm{JS}}\left(P||Q\right) & =D_{\mathrm{JS}}\left(Q||P\right)=\frac{1}{2}\left(D_{\mathrm{KL}}\left(P||M\right)+D_{\mathrm{KL}}\left(Q||M\right)\right)\\ \; & =\frac{1}{2}\sum_{i=1}^{n}\left(p_{i}\log_{2}\frac{2p_{i}}{p_{i}+q_{i}}+q_{i}\log_{2}\frac{2q_{i}}{p_{i}+q_{i}}\right) \end{array}$$
where *P* and *Q* are two PMFs associated with the same alphabet Z and *M* is the average distribution of *P* and *Q*. Each letter $z_{i}\in Z$ is associated with a probability value $p_{i}\in P$ and another $q_{i}\in Q$. With the base 2 logarithm as in Equation (Equation 8), $D_{\mathrm{JS}}\left(P||Q\right)$ is bounded by 0 and 1.

The square root of $D_{\mathrm{JS}}\left(P||Q\right)$, denoted as $\sqrt{D_{\mathrm{JS}}}\left(P||Q\right)$, is not only a bounded divergence measure, but also a distance metric [46,47]. It is thus interesting to include $\sqrt{D_{\mathrm{JS}}}$ as a candidate measure.

Another bounded measure is the conditional entropy H(P|Q):(9)$$H\left(P|Q\right)=H\left(P\right)-I\left(P;Q\right)=H\left(P\right)-\sum_{i=1}^{n}\sum_{j=1}^{n}r_{i,j}\log_{2}\frac{r_{i,j}}{p_{i}q_{j}}$$
where I(P;Q) is the mutual information between *P* and *Q* and $r_{i,j}$ is the joint probability of the two conditions of $z_{i},z_{j}\in Z$ that are associated with *P* and *Q*. H(P|Q) is bounded by 0 and H(P). Because I(P;Q) measures the amount of shared information between *P* and *Q* (and therefore a kind of similarity), H(P|Q) thus increases if *P* and *Q* are less similar. Note that we use H(P|Q) and I(P;Q) here in the context that *P* and *Q* are associated with the same alphabet Z, though the general definitions of H(P|Q) and I(P;Q) are more flexible.

The above two measures in Equations (Equation 8) and (Equation 9) consist of logarithmic scaling of probability values, in the same form of Shannon entropy. They are entropic measures. There are many other divergence measures in information theory, including many in the family of *f*-divergences [48]. However, many are also unbounded.

Meanwhile, entropic divergence measures belong to the broader family of statistical distances or difference measures. In this work, we considered a set of non-entropic measures in the form of Minkowski distances, which have the following general form:(10)$$D_{M}^{k}\left(P,Q\right)=\sqrt[{k}]{\sum_{i=1}^{n}{\left|p_{i}-q_{i}\right|}^{k}}\;\left(k>0\right)$$
where we use symbol *D* instead of D because it is not entropic.

### 5.3. New Candidates of Bounded Measures

For each letter $z_{i}\in Z$, $D_{\mathrm{KL}}\left(P||Q\right)$ measures the difference between its self-information $-\log_{2}\left(p_{i}\right)$ and $-\log_{2}\left(q_{i}\right)$ with respect to *P* and *Q*. Similarly, $D_{\mathrm{JS}}\left(P||Q\right)$ measures the difference of self-information with the involvement of an average distribution (P+Q)/2. Meanwhile, it will be interesting to consider the difference of two probability values, i.e., $|p_{i}-q_{i}|$, and the information content of the difference. This would lead to measuring $\log_{2}\left|p_{i}-q_{i}\right|$, which is unfortunately an unbounded term in [−∞,0].

Let $u=|p_{i}-q_{i}|$, the function $\log_{2}u^{k}+1$ (where k>0) is an isomorphic transformation of $\log_{2}u$. The former preserves all information of the latter, while offering a bounded measure in [0,1]. Although $\log_{2}u^{k}+1$ and $\log_{2}u$ are both monotonically increasing measures, they have different gradient functions, or visually, different shapes. We thus introduce a power parameter *k* to enable our investigation into different shapes. The introduction of *k* reflects the open-minded nature of this work. It follows the same generalization approach as Minkowski distances and α-divergences [49], avoiding a fixation on their special cases such as the Euclidean distance or $D_{\mathrm{KL}}$.

We first consider a commutative measure $D_{\mathrm{new}}^{k}$:(11)$$D_{\mathrm{new}}^{k}\left(P||Q\right)=\frac{1}{2}\sum_{i=1}^{n}\left(p_{i}+q_{i}\right)\log_{2}\left(|p_{i}-q_{i}{|}^{k}+1\right)$$
where k>0. Because $0\leq |p_{i}-q_{i}{|}^{k}\leq 1$, we have
$$\frac{1}{2}\sum_{i=1}^{n}\left(p_{i}+q_{i}\right)\log_{2}\left(0+1\right)\leq D_{\mathrm{new}}^{k}\left(P||Q\right)\leq \frac{1}{2}\sum_{i=1}^{n}\left(p_{i}+q_{i}\right)\log_{2}\left(1+1\right)$$
Since $\log_{2}1=0$, $\log_{2}2=1$, $\sum p_{i}=1$, $\sum q_{i}=1$, $D_{\mathrm{new}}^{k}\left(P||Q\right)$ is thus bounded by 0 and 1. The formulation of $D_{\mathrm{new}}^{k}\left(P||Q\right)$ was derived from its non-commutative version:(12)$$D_{\mathrm{ncm}}^{k}\left(P||Q\right)=\sum_{i=1}^{n}p_{i}\log_{2}\left(|p_{i}-q_{i}{|}^{k}+1\right)$$
which captures the non-commutative property of $D_{\mathrm{KL}}$. In this work, we focus on two options of $D_{\mathrm{new}}^{k}$ and $D_{\mathrm{ncm}}^{k}\left(P||Q\right)$, i.e., when k=1 and k=2.

As $D_{\mathrm{JS}}$, $D_{\mathrm{new}}^{k}$, and $D_{\mathrm{ncm}}^{k}$ are bounded by [0, 1], if any of them is selected to replace $D_{\mathrm{KL}}$, Equation (Equation 3) can be rewritten as
(13)$$\mathrm{Benefit}=H\left(Z_{i}\right)-H\left(Z_{i+1}\right)-H_{\max}\left(Z_{i}\right)D(Z_{i}^{\prime }\left|\right|Z_{i})$$
where $H_{\max}$ denotes maximum entropy, while D is a placeholder for $D_{\mathrm{JS}}$, $D_{\mathrm{new}}^{k}$, or $D_{\mathrm{ncm}}^{k}$. Note that while $H_{\max}\left(Z_{i}\right)D(Z_{i}^{\prime }\left|\right|Z_{i})$ is bounded by $H_{\max}\left(Z_{i}\right)$, $H_{\max}\left(Z_{i}\right)$ can have any non-negative value and is calculated as $\log_{2}∥Z_{i}∥$, where $∥Z_{i}∥$ is the number of letters in $Z_{i}$.

We have considered the option of using $H(Z_{i})$ instead of $H_{\max}\left(Z_{i}\right)$. However, this would lead to an undesirable paradox. Consider an alphabet $Z_{i}=\left\{z_{a},z_{b}\right\}$ with a PMF $P_{i}=\left\{p_{a},1-p_{a}\right\}$. Consider a simple visual mapping that is supposed to encode the probability value $p_{a}$ using the luminance of a monochrome shape with, $\mathrm{luminance}\left(p_{a}\right)=p_{a}$, black = 0, and white = 1. Unfortunately, the accompanying legend displays incorrect labels as black for $p_{a}=1$ and white for $p_{a}=0$. The visualization results thus feature a “lie” distribution $P_{i}=\left\{1-p_{a},p_{a}\right\}$. An obvious paradoxical scenario is when $P_{i}=\left\{1,0\right\}$, which has an entropy value $H(Z_{i})=0$. Although $D_{\mathrm{JS}}$, $D_{\mathrm{new}}^{k}$, and $D_{\mathrm{ncm}}^{k}$ would all return 1 as the maximum value of divergence for the visual mapping, the term $H\left(Z_{i}\right)D(Z_{i}^{\prime }\left|\right|Z_{i})$ would indicate that there would be no divergence. Hence $H(Z_{i})$ cannot be used instead of $H_{\max}\left(Z_{i}\right)$.

## 6. Conceptual Evaluation of Bounded Measures

Given those bounded candidates in the previous section, we would like to select the most suitable measure to be used in Equation (Equation 13). In the history of measurement science [50], there have been an enormous amount of research effort devoted to inventing, evaluating, and selecting different candidate measures (e.g., metric vs. imperial measurement systems; temperatute scales: Celsius, Fahrenheit, kelvin, Rankine, and Reaumur; and Seismic magnitude scales: Richter, Mercalli, moment magnitude, and many others). There is usually no ground truth as to which is correct, and the selection decision is rarely determined only by mathematical definitions or rules [51]. Similarly, there are numerous statistical distance and difference measures. selecting a measure in a certain application is often an informed decision based on multiple factors. Measuring the benefit of visualization and the related informative divergence in visualization processes is a new topic in the field of visualization. It is not unreasonable to expect that more research effort will be made in the coming years, decades, or unsurprisingly, centuries. The work presented in this two-part paper represents the early thought and early effort in this endeavor. In this work, we devised a set of criteria and conducted multi-criteria decision analysis (MCDA) [3] to evaluate the candidate measures described in the previous section.

Our criteria fall into two main categories. The first group of criteria reflect seven desirable conceptual or mathematical properties, as shown in Table 3. The second group of criteria reflect the assessments based on numerical instances constructed synthetically or obtained from experiments. This first part of the paper focuses conceptual evaluation based on the first group of criteria, while the second part focuses on empirical evaluation based on the second group of criteria [5].

For criteria 1, 6, and 7 in the first group, we use visualization plots to aid our analysis of the mathematical properties. Based on our analysis, we score each divergence measure against a criterion using ordinal values between 0 and 5 (0 unacceptable, 1 fall-short, 2 inadequate, 3 mediocre, 4 good, 5 best). We intentionally do not assign weights to these criteria. While we will offer our view as to the importance of different criteria, we encourage readers to apply their own judgement to weight these criteria. We hope that readers will reach the same conclusion as ours. We draw our conclusion about the conceptual evaluation in Section 7, where we also outline the need for data-driven empirical evaluation.

### 6.1. Criterion 1: Is It a Bounded Measure?

This is essential since the selected divergence measure is to be bounded. Otherwise we could just use the KL-divergence. Let us consider a simple alphabet $Z=\{z_{1},z_{2}\}$, which is associated with two PMFs, $P=\{p_{1},1-p_{1}\}$ and $Q=\{q_{1},1-q_{1}\}$. We set $q_{1}=\left(1-\alpha \right)p_{1}+\alpha \left(1-p_{1}\right),\alpha \in \left[0,1\right]$, such that when α=1, *Q* is most divergent away from *P*. The entropy values of *P* and *Q* fall into the range of [0, 1]. Hence semantically, it is more intuitive to reason an unsigned value representing their divergence within the same range.

Figure 5 shows several measures by varying the values of $p_{1}$ in the range of [0,1]. We can obverse that $D_{\mathrm{KL}}$ raises its values quickly above 1 when $\alpha =1,p_{1}\leq 0.22$. Its scaled version, $0.3D_{\mathrm{KL}}$, does not rise up as quick as $D_{\mathrm{KL}}$ but raises above 1 when $\alpha =1,p_{1}\leq 0.18$. In fact $D_{\mathrm{KL}}$ and $0.3D_{\mathrm{KL}}$ are not only unbounded, they do not return valid values when $p_{1}=0$ or $p_{1}=1$. We therefore score them 0 for Criterion 1.

$D_{\mathrm{JS}}$, $\sqrt{D_{\mathrm{JS}}}$, H(P|Q), $D_{\mathrm{new}}^{k}$, and $D_{\mathrm{ncm}}^{k}$ are all bounded by [0, 1] and they can potentially be used in the rewritten formula Equation (Equation 13). We score them 5. Although $D_{M}^{k}$ is a bounded measure, its semantic interpretation is not ideal, because its upper bound depends on *k* and is always >1. We thus score it 3. Although $0.3D_{\mathrm{KL}}$ is eliminated based on criterion 1, it is kept in Table 3 as a benchmark in analyzing criteria 2–7. Meanwhile, we carry all other scores forward to the next stage of analysis.

### 6.2. Criterion 2: How Many PMFs Does It Have as Dependent Variables

For criteria 2–7, we follow the base-criterion method [52] by considering $D_{\mathrm{KL}}$ and $0.3D_{\mathrm{KL}}$ as the benchmark. Criterion 2 concerns the number of PMFs as the dependent (or input) variables of each measure. $D_{\mathrm{KL}}$ and $0.3D_{\mathrm{KL}}$ depend on two PMFs, *P* and *Q*. All candidates of the bounded measures depend on two PMFs, except the conditional entropy H(P|Q) that depends on three. Because in most practical applications, it requires some effort to obtain a PMF, e.g., by observing an alphabet for a period. A joint probability distribution, which is required for calculating the mutual information term I(P;Q) in Equation (Equation 9), would need observation of both input and output alphabets of a process in a synchronized manner. The need for an extra PMF makes H(P|Q) much less favourable, and it is scored 2. All others are scored 5.

### 6.3. Criterion 3: Is It an Entropic Measure?

An *entropic* measure characteristically features a logarithmic transformation of some numerical compositions of probability values. The logarithmic transformation accentuates the change from a state of order to a state of disorder or vice versa. The probabilistic mean of such changes related to all letters in an alphabet pertains to the Shannon entropy. With base 2 logarithm, an entropic measure usually has or features the unit bit. As the AC term in Equation (Equation 3) and the original PD term (i.e., $D_{\mathrm{KL}}$) are measured in bits, we prefer to have an entropic divergence measure for the PD term so the “benefit” can be measured in bits. For this reason, $D_{M}^{k}$ is scored 1, and all others are given 5.

### 6.4. Criterion 4: Is It a Distance Measure?

When a measure is referred to as a divergence measure, it usually implies that it is not a distance metric. A true distance metric must have the following mathematical properties:identity: d(x,y)=0⇔x=y,symmetry: d(x,y)=d(y,z),triangle inequality: d(x,y)≤d(x,z)+d(z,y),non-negativity: d(x,y)≥0.

The first three conditions are axioms of a metric system. Among the candidate measures, $D_{M}^{k}$ and $\sqrt{D_{\mathrm{JS}}}$ are metrics. They are scored 5. $0.3D_{\mathrm{KL}}$, H(P|Q), $D_{\mathrm{ncm}}^{k=1}$, and $D_{\mathrm{ncm}}^{k=2}$ satisfy only conditions 1 and 4. They are scored 2. $D_{\mathrm{JS}}$ and $D_{\mathrm{new}}^{k=2}$ satisfy conditions 1, 2, 4, and they are scored 3. At the moment, we do not know if $D_{\mathrm{new}}^{k=1}$ is a metric or not. There is a mathematical proof to show that $D_{\mathrm{new}}^{k=1}$ is a metric for 2-letter alphabets [53], but a proof or disproof for *n*-letter alphabets is yet known. We thus give $D_{\mathrm{new}}^{k=1}$ a score 4.

### 6.5. Criterion 5: Is It Intuitive or Easy to Understand?

One reason that $D_{\mathrm{KL}}$ is the most popular divergence measure is that it is easy to understand the meaning of its element function f(p,q)=log(p/q)=log(p)−log(q), where p,q∈[0,1] are two probability values. It is the difference of the logarithmic representations of *p* and *q*. As $0.3D_{\mathrm{KL}}$ introduces a global scaling transformation, it adds a barrier in understanding. We take one score away for such a barrier by giving $0.3D_{\mathrm{KL}}$ a score 4.

As shown in Equation (Equation 8), $D_{\mathrm{JS}}$ introduces an intermediate value m=(p+q)/2. The element function splits into two parts f(p,m) and f(q,m). Such a transformation adds a barrier in our appreciation of meaning of the measure. Note that one could take two points away as this transformation is rather complex. Nevertheless, for consistency, we take one point away per transformation. $\sqrt{D_{\mathrm{JS}}}$ introduces a square root as a global transformation, adding a further barrier in understanding. With $D_{\mathrm{KL}}$ as the benchmark (score 5), we score $D_{\mathrm{JS}}$ 4 and $\sqrt{D_{\mathrm{JS}}}$ 3 by counting the number of barriers in understanding.

Consider another element function g(p,q)=log|p−q|, which is the logarithmic representation of the difference between *p* and *q*. It is as easy to understand as f(p,q). $D_{\mathrm{new}}^{k=1}$ introduces a transformation as log(|p−q|+1), while $D_{\mathrm{new}}^{k=2}$ introduces an additional one as $\log(|p-q{|}^{2}+1)$. Each transformation adds a new barrier in understanding. We therefore give $D_{\mathrm{new}}^{k=1}$ a score 4 and $D_{\mathrm{new}}^{k=2}$ a 3. Similarly we assign a score 4 to $D_{\mathrm{ncm}}^{k=1}$ and a 3 to $D_{\mathrm{ncm}}^{k=2}$.

For *n*-letter alphabet, $D_{M}^{k}\left(k=2\right)$ is the same as the *n*-dimensional Euclidean distance. We thus gives it a full score 5. As $D_{M}^{k}\left(k=200\right)$ is considered to have an extra barrier in understanding, we score it 4.

Finally, H(P|Q) is the composition of two commonly-used information-theoretic measures. We give it a score 4 by considering the composition as a transformation.

### 6.6. Criterion 6: Visual Analysis of Curve Shapes in the Range of (0,1)

One may wish for a bounded measure to have a geometric behaviour similar to $D_{\mathrm{KL}}$ since it is the most popular divergence measure. Since $D_{\mathrm{KL}}$ rises up far too quickly as shown in Figure 5, we use $0.3D_{\mathrm{KL}}$ as a benchmark, though it is still unbounded. As Figure 5 plots the curves for α=0.0,0.1,…,1.0, we can visualize the “geometric shape” of each bounded measure, and compare it with that of $0.3D_{\mathrm{KL}}$.

From Figure 5, we can observe that $D_{\mathrm{JS}}$ has almost a perfect match when α=0.5, while $D_{\mathrm{new}}^{k}\left(k=2\right)$ is also fairly close. They thus score 5 and 4 respectively in Table 3. Meanwhile, the lines of H(P|Q) curve in the opposite direction of $0.3D_{\mathrm{KL}}$. We score it 1. $\sqrt{D_{\mathrm{JS}}}$, $D_{\mathrm{new}}^{k}\left(k=1\right)$, and $D_{M}^{k}\left(k=2,k=200\right)$ are of similar shapes. In terms of the direction of curvature, $\sqrt{D_{\mathrm{JS}}}$ correlates slightly better with $0.3D_{\mathrm{KL}}$ than $D_{M}^{k}$ and $D_{\mathrm{new}}^{k}\left(k=1\right)$. We thus assign a score 3 to $\sqrt{D_{\mathrm{JS}}}$ and a score 2 to $D_{\mathrm{new}}^{k}\left(k=1\right)$ and $D_{M}^{k}\left(k=2,k=200\right)$. For the PMFs *P* and *Q* concerned, $D_{\mathrm{ncm}}^{k}$ has the same curves as $D_{\mathrm{new}}^{k}$. Hence $D_{\mathrm{ncm}}^{k}$ has the same score as $D_{\mathrm{new}}^{k}$ in Table 3.

### 6.7. Criterion 7: Visual Analysis of Curve Shapes in a Range near Zero, i.e., $\left[0.1^{10},0.1\right]$

We now consider Figure 6, where the candidate measures are visualized in comparison with $D_{\mathrm{KL}}$ and $0.3D_{\mathrm{KL}}$ in a range close to zero, i.e., $\left[0.1^{10},0.1\right]$. The ranges $\left[0,0.1^{10}\right]$ and [0.1,0.5] are there only for references to the nearby contexts as they do not have the same logarithmic scale as that in the range $\left[0.1^{10},0.1\right]$. We can observe that in $\left[0.1^{10},0.1\right]$ the curve of $0.3D_{\mathrm{KL}}$ rises as almost quickly as $D_{\mathrm{KL}}$. This confirms that simply scaling the KL-divergence is not an adequate solution. The curves of $D_{\mathrm{new}}^{k=1}$ and $D_{\mathrm{new}}^{k=2}$ converge to their maximum value 1.0 earlier than that of $D_{\mathrm{JS}}$. The $\sqrt{D_{\mathrm{JS}}}$ curve appears between those of $D_{\mathrm{new}}^{k=1}$ and $D_{\mathrm{new}}^{k=2}$. If the curve of $0.3D_{\mathrm{KL}}$ is used as a benchmark as in Figure 5, the curve of $D_{\mathrm{new}}^{k=2}$ is much closer to $0.3D_{\mathrm{KL}}$ than that of $D_{\mathrm{JS}}$. We thus score $D_{\mathrm{new}}^{k=2}$: 5, $\sqrt{D_{\mathrm{JS}}}$: 4, $D_{\mathrm{JS}}$: 3, $D_{\mathrm{new}}^{k=1}$: 3, $D_{M}^{k}\left(k=200\right)$: 3, $D_{M}^{k}\left(k=200\right)$: 2, and H(P|Q): 1. Same as Figure 5, $D_{\mathrm{ncm}}^{k}$ has the same curves and thus the same score as $D_{\mathrm{new}}^{k}$.

The sums of the scores for criteria 1–7 indicate that H(P|Q) and $D_{M}^{k}$ are much less favourable than $D_{\mathrm{JS}}$, $\sqrt{D_{\mathrm{JS}}}$, $D_{\mathrm{new}}^{k}$, and $D_{\mathrm{ncm}}^{k}$. Because these criteria have more holistic significance than the data-driven analysis in the second part of this paper [5], we can eliminate H(P|Q) and $D_{M}^{k}$ for further consideration. Ordinal scores in MCDA are typically subjective. Nevertheless, in our analysis, ±1 in those scores would not affect the elimination.

## 7. Discussions and Conclusions

In this paper, we have considered the need to improve the mathematical formulation of an information-theoretic measure for analyzing the cost–benefit of visualization as well as other processes in a data intelligence workflow [1]. The concern about the original measure is its unbounded term based on the KL-divergence. As discussed in the early sections of this paper, although using the KL-divergence measure in [1] as part of the cost–benefit measure is a conventional or orthodox choice, its unboundedness leads to several issues in the potential applications of the cost–benefit measure to practical problems:It is not intuitive to interpret a set of values that would indicate that the amount of distortion in viewing a visualization that features some information loss, could be much more than the total amount of information contained in the visualization.It is difficult to specify some simple visualization phenomena. For example, before a viewer observes a variable *x* using visualization, the viewer incorrectly assumes that the variable is a constant (e.g., x≡10, and probability p(10)=1). The KL-divergence cannot measure the potential distortion of this phenomenon of bias because this is a singularity condition, unless one changes p(10) by subtracting a small value 0<ϵ<1.If one tries to restrict the KL-divergence to return values within a bounded range, e.g., determined by the maximum entropy of the visualization space or the underlying data space, one could potentially lose a non-trivial portion of the probability range (e.g., 13% in the case of a binary alphabet).

To address these problems, we have proposed to replace the KL-divergence in the cost–benefit measure with a bounded measure. We have obtained a proof that the divergence used in the cost–benefit formula is conceptually bounded, as long as the input and output alphabets of a process have a finite number of letters.

We have considered a number of bounded measures to replace the unbounded term, including a new divergence measure $D_{\mathrm{new}}^{k}$ and its variant $D_{\mathrm{ncm}}^{k}$. We have conducted multi-criteria decision analysis to select the best measure among these candidates. In particular, we have used visualization to aid the observation of the mathematical properties of the candidate measures, assisting in the analysis of three criteria in considered in this paper.

From Table 3, we can observe the process of narrowing down from eight candidate measures to five measures. In particular, three candidate measures $D_{\mathrm{JS}}$, $\sqrt{D_{\mathrm{JS}}}$, and $D_{\mathrm{new}}^{k=2})$ received the same total scores. They are followed by $D_{\mathrm{ncm}}^{k=2}$ and $D_{\mathrm{new}}^{k=1}$. It is not easy to separate them. We therefore conducted two groups of case studies to collect empirical evidence for further evaluating these candidate measures.

In the history of measurement science [50], as shown in Figure 7, scientists encountered many similar dilemma in choosing different measures. For example, temperature measures Celsius, Fahrenheit, Réaumur, Rømer, and Delisle scales exhibit similar mathematical properties, their proposal and adoption were largely determined by practical instances:Rømer—0 degree: freezing brine, 7.5 degree: the freezing point of water, 60 degree: the boiling point of water;Fahrenheit (original)—0 degree: the freezing point of brine (a high-concentration solution of salt in water), 32 degree: ice water, 96 degree: average human body temperature;Fahrenheit (present)—32 degree: the freezing point of water, 212 degree: the boiling point of water;Réaumur—0 degree: the freezing point of water, 80 degree: the boiling point of water;Delisle—0 degree: the boiling point of water, −1 degree; the contraction of the mercury in hundred-thousandths.Celsius* (original)—0 degree: the boiling point of water, 100 degree: the freezing point of water;Celsius (1743–1954)—0 degree: the freezing point of water, 100 degree: the boiling point of water;Celsius (1954–2019)—redefined based on absolute zero and the triple point of VSMOW (specially prepared water);Celsius (2019–now)—redefined based on the Boltzmann constant.

Before the development of these scales, Newton proposed two temperature systems based on his observation of some 18 instance values [54]. The effort of these scientific pioneers suggested that observing how candidate measures relate to practical instances was part of the scientific processes for selecting different candidate measures.

The work reported in this paper outlines a conceptual notion that measuring the divergence that may be caused by a transformation from an input alphabet (e.g., data) to an output alphabet (e.g., visualization) is more intuitive if it has a lower bound 0 and an upper bound of the maximum entropy of the input alphabet (as enforced in Equation (Equation 13)). The more complex the input alphabet (i.e., the input information space), the wider the range of the potential divergence. As all candidate measures are bounded by [0, 1] and the maximum entropy of an alphabet is easy to calculate, we have addressed part of the problem where $D_{\mathrm{KL}}$ has no upper bound regardless how complex the input alphabet is.

Building on the work presented in this paper, we carried out further investigation into a group of criteria based on observed instances in synthetic and experimental data. This data-driven evaluation is presented in the second part of this paper [5], where we aim to narrow the remaining six candidate measures to one measure, and to revise the original cost–benefit ratio in [1] based on the combined conclusion derived from the conceptual evaluation (i.e., this work) and empirical evaluation. The empirical evidence collected in several case studies helps identify some additional strengths and weaknesses of the remaining six candidate measures. Based on conceptual and empirical criteria considered in both parts of the paper, we will offer a conclusion that the candidate measure $D_{\mathrm{new}}^{k}\left(k=2\right)$ is ahead of $\sqrt{D_{\mathrm{JS}}}$, especially when we include an additional conceptual criterion discovered during the case studies. Readers can find the detailed description and analysis of these case studies in the second part of this paper [5].

## Figures and Tables

**Figure 1 entropy-24-00228-f001:**
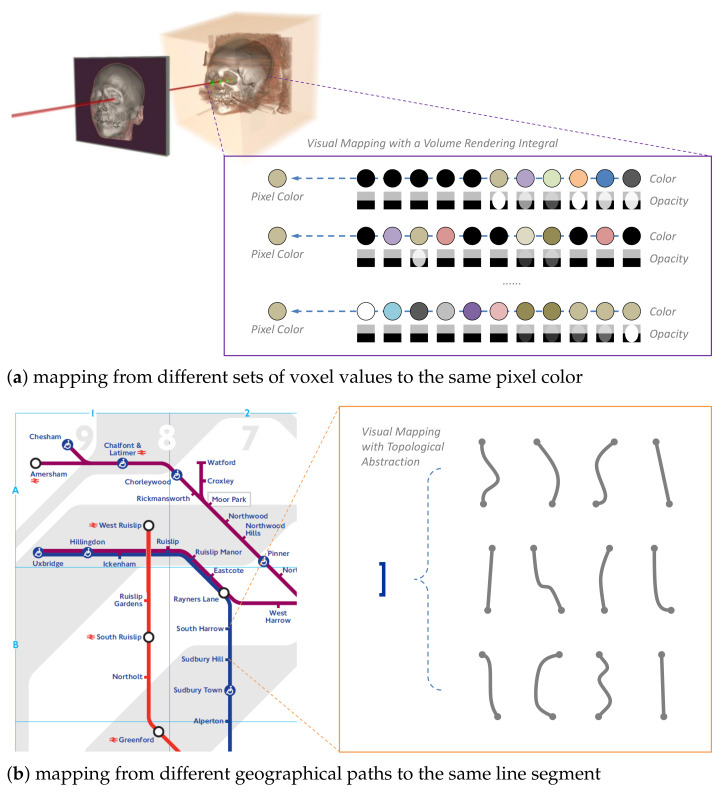
Visual encoding typically features many-to-one mapping from data to visual representations, hence information loss. For example, (**a**) in volume visualization, the color of each pixel results from a complex process of combining a sequence of voxel values, and (**b**) in metro maps, different geographical paths are often represented using indistinguishable line segments. The significant amount of information loss in volume visualization and metro maps suggests that viewers not only can abide the information loss but also benefit from it. Measuring such benefits can lead to new advancements of visualization, in theory and practice.

**Figure 2 entropy-24-00228-f002:**
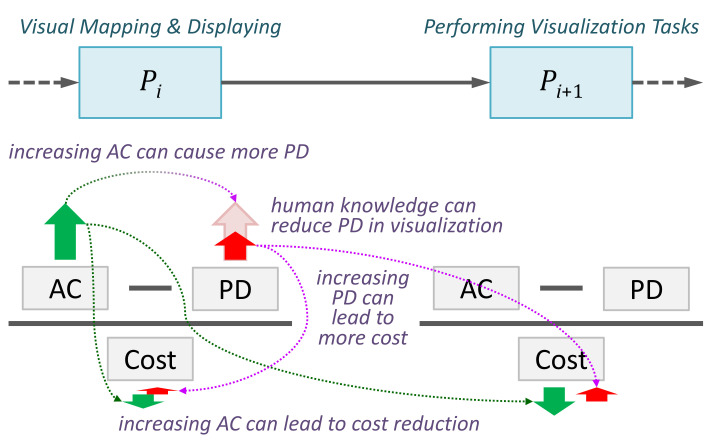
Each process in a data intelligence workflow can be characterized using three abstract measures: *alphabet compression* (AC), *potential distortion* (PD), and *cost*. They can be used to reason about the shortcomings in a workflow and identify possible solutions in abstraction [37]. For example, increasing data filtering in visualization (AC) may reduce the cost of $P_{i}$ and $P_{i+1}$, especially when human knowledge can reduce perceptual errors (PD).

**Figure 3 entropy-24-00228-f003:**
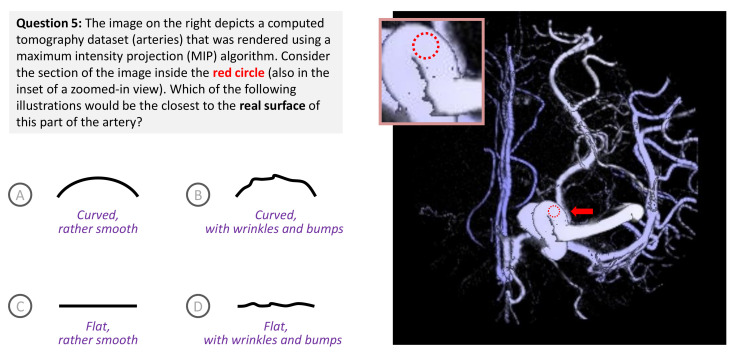
A volume dataset was rendered using the maximum intensity projection (MIP) method, which causes curved surfaces of arteries to appear rather flat. Posing a question about a “flat area” in the image can be used to tease out a viewer’s knowledge that is useful in a visualization process.

**Figure 4 entropy-24-00228-f004:**
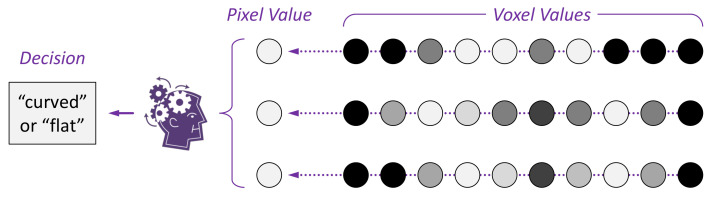
In this 2D illustration of a simplified scenario of volume visualization, three sequences of voxels are rendered using the MIP method. The volume on the right features a curved surface defined by those brightest voxels. By projecting the maximum voxel values to the pixels in the middle, the curvature information of the surface is lost. A viewer needs to determine if the surface in the volume is curved or flat, for which the viewer’s knowledge is critical.

**Figure 5 entropy-24-00228-f005:**
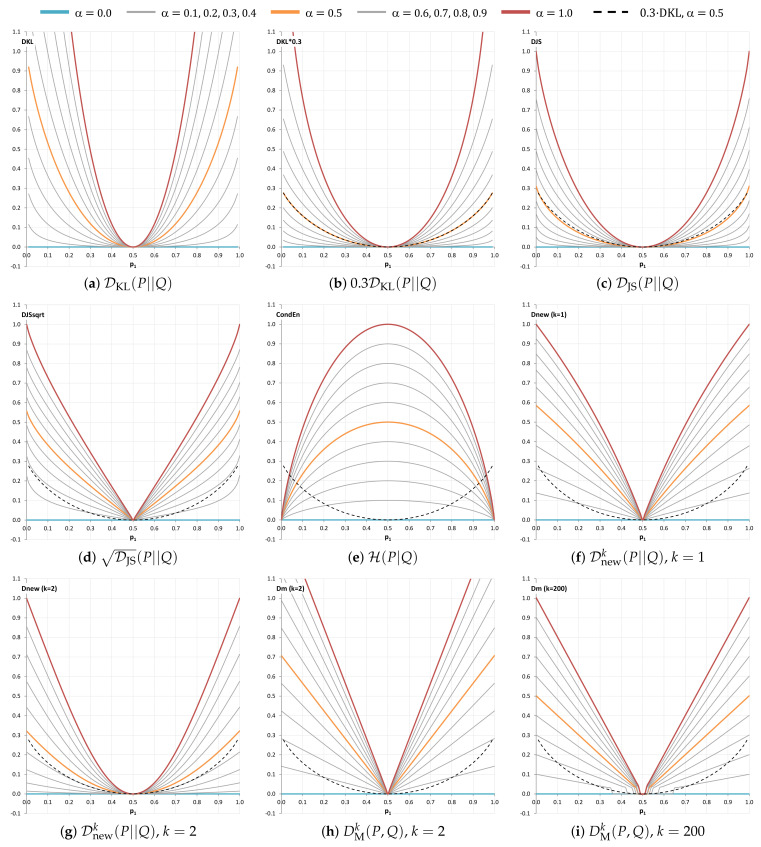
The different measurements of the divergence of two PMFs, $P=\{p_{1},1-p_{1}\}$ and $Q=\{q_{1},1-q_{1}\}$. The *x*-axis shows $p_{1}$, varying from 0 to 1, while we set $q_{1}=\left(1-\alpha \right)p_{1}+\alpha \left(1-p_{1}\right),\alpha \in \left[0,1\right]$. When α=1, *Q* is most divergent away from *P*. The curve $0.3D_{\mathrm{KL}}\left(\alpha =0.5\right)$ is shown in a dashed black line, and is used as a benchmark for observing the corresponding curves (in orange) produced by the candidate measures in (**c**–**i**).

**Figure 6 entropy-24-00228-f006:**
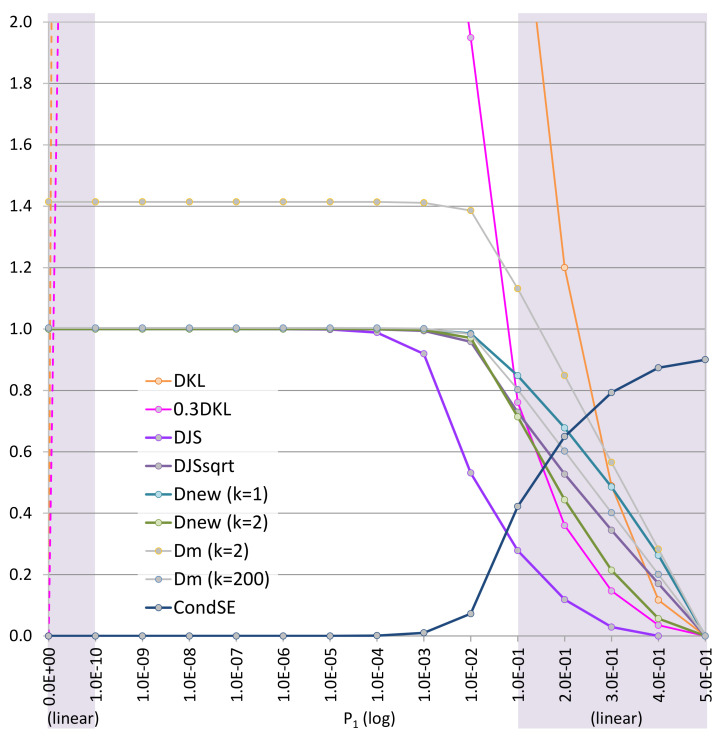
A visual comparison of the candidate measures in a range near zero. Similar to Figure 5, $P=\{p_{1},1-p_{1}\}$ and $Q=\{q_{1},1-q_{1}\}$, but only the curve α=1 is shown, i.e., $q_{1}=1-p_{1}$. The line segments of $D_{\mathrm{KL}}$ and $0.3D_{\mathrm{KL}}$ in the range $\left[0,0.1^{10}\right]$ do not represent the actual curves. The ranges $\left[0,0.1^{10}\right]$ and [0.1,0.5] are only for references to the nearby contexts as they do not use the same logarithmic scale as in $\left[0.1^{10},0.1\right]$.

**Figure 7 entropy-24-00228-f007:**
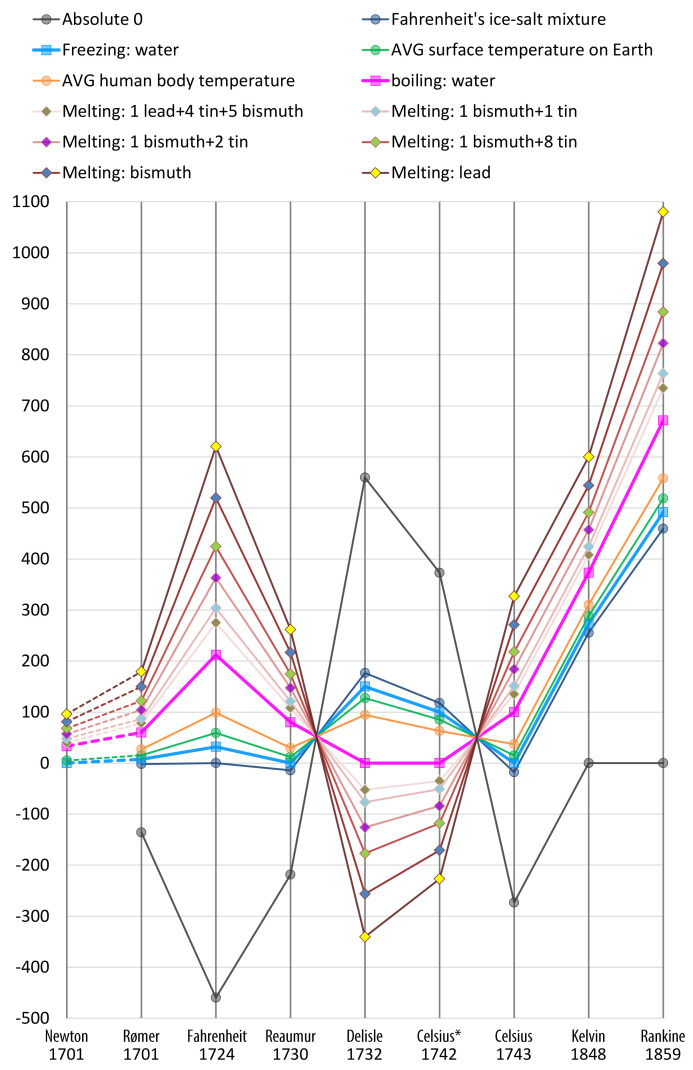
Some of the major temperature scales considered by scientists in the history. It took four decades from Isaac Newton’s instance-based proposal to arrive at the most-commonly used Celsius scale. It took another century to discover absolute zero as the lower bound.

**Table 1 entropy-24-00228-t001:** Imaginary scenarios where probability data is collected for estimating knowledge related to alphabet A={curved,flat}. The ground truth (G.T.) PMFs are defined with ϵ=0.01 and 0.0001 respectively. The potential distortion (as “→ value”) is computed using the KL-divergence.

	Scenario 1	Scenario 2
$Q(A_{G.T.})$:	{0.99,0.01}	{0.9999,0.0001}
$P(A_{\mathrm{MIP}})$:	{0.01,0.99}→6.50	{0.0001,0.9999}→13.28
$P(A_{\mathrm{doctors}})$:	{0.99,0.01}→0.00	{0.99,0.01}→0.05
$P(A_{\mathrm{patients}})$:	{0.7,0.3}→1.12	{0.7,0.3}→3.11

**Table 2 entropy-24-00228-t002:** Imaginary scenarios for estimating knowledge related to alphabet B={wrinkles−and−bumps,smooth}. The ground truth (G.T.) PMFs are defined with ϵ=0.1 and 0.001 respectively. The potential distortion (as “→ value”) is computed using the KL-divergence.

	Scenario 3	Scenario 4
$Q(B_{G.T.})$:	{0.9,0.1}	{0.999,0.001}
$P(B_{\mathrm{MIP}})$:	{0.1,0.9}→2.54	{0.001,0.999}→9.94
$P(B_{\mathrm{doctors}})$:	{0.8,0.2}→0.06	{0.8,0.2}→1.27
$P(B_{\mathrm{patients}})$:	{0.1,0.9}→2.54	{0.1,0.9}→8.50

**Table 3 entropy-24-00228-t003:** A summary of multi-criteria decision analysis in the first part of this paper. Each measure is scored against a conceptual criterion using an integer in [0, 5] with 5 being the best. The symbol ▸ indicates an interim conclusion after considering one or a few criteria. In the second part of the paper [5], we will discuss another five criteria.

Criteria	Importance	$0.3D_{\mathrm{KL}}$	$D_{\mathrm{JS}}$	$\sqrt{D_{\mathrm{JS}}}$	H(P|Q)	$D_{\mathrm{new}}^{k=1}$	$D_{\mathrm{new}}^{k=2}$	$D_{\mathrm{ncm}}^{k=1}$	$D_{\mathrm{ncm}}^{k=2}$	$D_{M}^{k=2}$	$D_{M}^{k=200}$
1. Boundedness	critical	0	5	5	5	5	5	5	5	3	3
▸*$0.3D_{\mathrm{KL}}$ is eliminated but used below only for comparison. The other scores are carried forward.*											
2. Number of PMFs	important	5	5	5	2	5	5	5	5	5	5
3. Entropic measures	important	5	5	5	5	5	5	5	5	1	1
4. Distance metric	helpful	2	3	5	2	4	3	2	2	5	5
5. Easy to understand	helpful	4	4	3	4	4	3	4	3	5	4
6. Curve shapes (Figure 5)	helpful	5	5	3	1	2	4	2	4	2	2
7. Curve shapes (Figure 6)	helpful	5	3	4	1	3	5	3	5	2	3
▸*Eliminate*H(P|Q), $D_{M}^{2}$, $D_{M}^{200}$*based on criteria 1–7*		**sum:**	**30**	**30**	**20**	**28**	**30**	**26**	**29**	**23**	**23**

## Data Availability

Not applicable.

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
