# Peer review of "A Bounded Measure for Estimating the Benefit of Visualization (Part I): Theoretical Discourse and Conceptual Evaluation"

_entropy, 2022, doi:10.3390/e24020228_

Round 1

Reviewer 1 Report

Summary:   

In this paper, the authors proposed to revise the existing cost-benefit measure by replacing the unbounded term with a bounded one. The authors focused on the conceptual analysis of the mathematical properties of these candidate measures and used visualization to support the multi-criteria comparison, narrowing the search down to several options with better mathematical properties. The theoretical discourse and conceptual evaluation in this part provide the basis for further data-driven evaluation based on synthetic and experimental case studies.

Comments:

1 . My main concern towards this paper is that the evaluation is conceptual only. There are no much we can do to tell the true advantage of the proposed new bounded measures.  The seven criterions listed in the paper, are indeed reasonable, however, they are not really convincing to the readers that any one of them can be the golden standard to choosing the measure. 

  1. For the same reason as above, after reading the paper, I still have no clue on the final conclusions that which measure is better?

  1. For properties such as boundedness, although empirical evaluations are helpful, I think it is more convincing to directly derive the bound. 

  1. The scoring metric may need more careful justifications and careful design (like the visual analysis seems quite casual). The current scoring is too conceptual and less scientific.

Author Response

See the attached PDF file.

Reviewer 2 Report

The paper seems to contain interesting material, and the application of the idea in this work could be important.  However, before a possible publication, some changes are required. Please following my suggestions below.

- The degree of novelty should be clarified. Can you add a list or a table with examples  of measures suggested? in a more clear way than in your Table 3 (giving also the formulas etc.).

- Please improve the captions of all the Figures in the work, adding more explanations.

- In order to increase the range of possible applications (and  maybe improve the state-of-the-art discussion), please discuss if your measures can be applied in a importance sampling context as 

L. Martino, V. Elvira, "Effective Sample Size Approximations as Entropy Measures", viXra:2111.0145, 2021,

A. Kong. A note on importance sampling using standardized weights. Technical Report 348, Department of Statistics, University of Chicago, 1992.

There is also another interesting reference in the journal Signal Processing (related to the first paper above) that can give other interesting measure expressions. Please, discuss possible relationships and differences with  your work. Some of them are similar to some examples given in your Table 3.

Author Response

See the attached PDF file.
